# Purchasing Behavior, Setting, Pricing, Family: Determinants of School Lunch Participation

**DOI:** 10.3390/nu13124209

**Published:** 2021-11-24

**Authors:** Carolin Sobek, Peggy Ober, Sarah Abel, Ulrike Spielau, Wieland Kiess, Christof Meigen, Tanja Poulain, Ulrike Igel, Mandy Vogel, Tobias Lipek

**Affiliations:** 1LIFE Child, Hospital for Children and Adolescents, Medical Faculty, Leipzig University, 04103 Leipzig, Germany; Peggy.Ober@medizin.uni-leipzig.de (P.O.); Sarah.Abel@medizin.uni-leipzig.de (S.A.); Wieland.Kiess@medizin.uni-leipzig.de (W.K.); Christof.Meigen@medizin.uni-leipzig.de (C.M.); Tanja.Poulain@medizin.uni-leipzig.de (T.P.); ulrike.igel@fh-erfurt.de (U.I.); Mandy.Vogel@medizin.uni-leipzig.de (M.V.); Tobias.Lipek@medizin.uni-leipzig.de (T.L.); 2Center for Pediatric Research (CPL), Hospital for Children and Adolescents, Medical Faculty, Leipzig University, 04103 Leipzig, Germany; Ulrike.Spielau@medizin.uni-leipzig.de; 3Integrated Research and Treatment Center Adiposity Diseases, Medical Faculty, Leipzig University, 04103 Leipzig, Germany; 4Department of Social Work, University of Applied Science, 99085 Erfurt, Germany

**Keywords:** school lunch participation, food habits, social inequalities, overweight, children

## Abstract

Despite growing school lunch availability in Germany, its utilization is still low, and students resort to unhealthy alternatives. We investigated predictors of school lunch participation and reasons for nonparticipation in 1215 schoolchildren. Children reported meal habits, parents provided family-related information (like socioeconomic status), and anthropometry was conducted on-site in schools. Associations between school lunch participation and family-related predictors were estimated using logistic regression controlling for age and gender if necessary. School was added as a random effect. School lunch participation was primarily associated with family factors. While having breakfast on schooldays was positively associated with school lunch participation (OR_adj_ = 2.20, *p* = 0.002), lower secondary schools (OR_adj_ = 0.52, *p* < 0.001) and low SES (OR_adj_ = 0.25, *p* < 0.001) were negatively associated. The main reasons for nonparticipation were school- and lunch-related factors (taste, time constraints, pricing). Parents reported pricing as crucial a reason as an unpleasant taste for nonparticipation. Nonparticipants bought sandwiches and energy drinks significantly more often on school days, whereas participants were less often affected by overweight (OR = 0.66, *p* = 0.043). Our data stress school- and lunch-related factors as an important opportunity to foster school lunch utilization.

## 1. Introduction

In Germany, elementary school students spend more than 20 h, and secondary school students spend more than 30 h per week at school [1]. During this time, the school is in charge of the students’ well-being beyond mere teaching. Breakfast and lunch provision was traditionally ensured by the family, and the meals were served at or brought from home, but the traditional patterns are changing. However, healthy nutrition is essential to maintain or improve students’ health and subsequently enables or enhances academic performance [2,3]. 

In Germany, schooling was traditionally associated with an academic curriculum taught between about eight o’clock in the morning and—at least in primary school—noon or one o’clock in the afternoon. Hobbies, games, and other extracurricular activities are not generally offered and lunch is not provided [4]. Fostered by the expansion of the all-day school system with homework assistance and after-school programs in more and more German schools, there was a need for lunch provision during school time, and subsequently, school lunch availability doubled during the last decade [5,6]. All-day schools were even obliged to offer lunch by the Standing Conference of Ministers of Education and Cultural Affairs since 2006 [7]. 

The number of all-day schools increased from 4951 (16%) in 2002 to 13,381 (48%) in 2009, and further to 18,948 (71%) in 2019 [7]. Additionally, upper secondary education was shortened by one school year (12 instead of 13 years) while keeping the overall number of instructional hours and the duration of school vacation unchanged [8]. Therefore, the daily number of instructional hours increased. Further, an increasing number of families cannot provide meals during school days in a society increasingly dominated by shared earnings relationships, mainly associated with higher employment rates of mothers [9,10,11,12,13]. Therefore, the share of children dependent on out-of-home food provision is increasing. Moreover, substantial inequities in dietary environments across schools [14] can lead to unfavorable eating behavior and overweight [15,16], especially in socially disadvantaged neighborhoods. Such unhealthy eating behavior may also be avoidable by promoting school lunch participation.

Hence, the purpose of school lunch goes beyond satisfying hunger. It is an opportunity to offer healthy food and foster healthy nutrition habits that cannot be ignored, more so in the context of nutrition-related diseases like childhood obesity, with the still high prevalence [17], particularly among children with low SES [18]. 

In summary, although the availability of school lunch grew in recent years, utilization is still low. In previous research, school lunch participation factors were associated with school-related factors, like schedule or the school neighborhood [6,19], and with the lunch itself (pricing, quality, variety, or taste). The third group of factors are personal or family-related conditions and behaviors (special diet, family eating behavior, disadvantaged background). Whereas the latter are elusive to interventions requiring willingness and endurance from the participants and their families, providing free school meals [2] or boosting the students’ and parents’ positive perception (healthy, timely, high-quality) of the school lunch might be effective [20,21,22]. These intervention approaches have a positive short-term impact on the students’ health and well-being and have substantial long-term effects on students, particularly those from disadvantaged households [23]. Therefore, we investigated the current utilization rates in a German city and the associated factors, including both the parents’ and the students’ perspectives. Besides, we investigated associations between school lunch participation and students’ weight status.

## 2. Materials and Methods

### 2.1. Study Design and Population

Data were collected within the cross-sectional Leipzig School Nutrition Study [24]. The study was conducted in public and private schools chosen from predefined areas in Leipzig, Germany. The areas were selected based on socioeconomic data and overweight prevalence described by Igel et al. [25]. Data collection took place between May 2018 and May 2019. After the local education authority approved the study, all 42 eligible schools were invited to participate, of which 34 agreed to participate. All fourth graders (elementary schools equivalent to ISCED1 (According to the UNESCO International Standard Classification of Education [26])) and grades 6–8 (lower resp. upper secondary schools equivalent to ISCED2 resp. ISCED3) were invited, written informed consent was obtained from parents of 1215 participating children and adolescents (see Figure 1 for rates). 

The schools did not receive any financial benefits. After participating and returning the parent questionnaire, the children received an incentive of five euros. The Ethical Committee of the Medical Faculty of the Leipzig University approved the Leipzig School Nutrition Study (number 483/17-ek). Further, the study is registered with the German Clinical Trials Register (DRKS00017317).

### 2.2. Data

Anthropometric measurements and the completion of the children’s questionnaires took place in the schools. Parents’ questionnaires were completed at home and returned within one week after the on-site examination (Figure 1).

Bodyweight was measured wearing underwear and one layer of top clothing to the nearest 0.1 kg using a calibrated electronic scale (Kern, Balingen, Germany). Subsequently, weight was adjusted for the approximate weight of the clothing [27,28]. Body height was measured without wearing shoes to the nearest 0.1 cm, using a portable stadiometer (Seca, Hamburg, Germany). The body mass index (BMI) was calculated and transformed to standard deviation scores (SDS, equivalent to Z-scores) according to the German guidelines of the German Working Group of Obesity in Childhood and Adolescents [29] using German standard references [30]. Accordingly, the weight groups were defined as normal weight (BMI-SDS < 1.28) and overweight/obese (BMI-SDS ≥ 1.28) [29,30]. Trained pediatric study assistants conducted all procedures following standard protocols.

Eating behavior was assessed using a questionnaire completed by the children themselves, respective items are shown in Table 1. The students were asked if they ate at the school canteen. Children who checked the “yes” option were defined as school lunch participants. Accordingly, children who checked the “no” option were defined as non-participants.

For the analysis of seasonal variation in purchasing behavior, summer was defined as the period from May to October (Classification based on the mean monthly temperatures in the survey period in Saxony, Germany [31]). 

Parents completed a questionnaire with school-lunch related items, displayed in Table 2. Questions regarding unfavorable family eating behavior (FEB) were: “Does your child usually eat dinner together with the family?” (yes = 0/no = 1), “Is the TV usually running at home during dinner, or is a tablet, smartphone, cell phone, or similar being used?” (yes = 1/no = 0) and “Does your child usually snack between meals (e.g., chocolate, gummy bears, potato chips, pretzel sticks)?” (yes = 1/no = 0). The respective score was calculated as a sum of the three questions’ scores (if the sum was ≥2, the family’s eating behavior was defined as “unfavorable”). Classification into “unfavorable” was based on the current literature.

The socioeconomic index (modified Winkler Index [32]) combined information on the parents’ highest level of education (general and vocational), the current occupational position, and monthly equivalized disposable household income. The composite sum score ranges between three and 21 and can be classified into low, medium, and high [33]. In addition, the migration background was assessed. The respective dichotomous variable was defined as “Yes” if at least one parent was born abroad. Besides, parents were asked how many siblings their children had. Families with more than two siblings were defined as families with many children. Self-reported weight and height of both parents were used to calculate parental BMI values. Subjects with a BMI ≥ 25 kg/m^2^ were classified as overweight following the WHO definition [34]. A dichotomous variable, parental overweight, was defined (“Yes” if at least one parent was overweight).

### 2.3. Statistical Analysis

Descriptives are given as the mean and standard deviation for continuous variables and counts and percentages for categorical variables. 

Bivariate associations between participation in school lunch as outcome and independent variables were estimated using logistic regression, adjusting for age and gender if necessary. Age, gender, school type, BMI-SDS, SES group, migration background, parental overweight, many children, FEB, having breakfast on schooldays, purchasing, and bringing something were included as independent variables.

Finally, a multivariate hierarchical logistic regression analysis was performed with overweight/obese as dependent and school lunch participation as independent variables. Age, gender, school type, SES group, migration background, parental overweight, and having breakfast were included as covariates. Correspondingly, the multivariate hierarchical linear regression models were used to assess the associations between BMI-SDS and the covariates. Final models were determined based on the whole model using stepwise backward deletion.

The school was added as a random effect to account for clustering effects in all hierarchical models except models only containing school type as a covariate because of a strong dependence between school type and the school itself. Due to high collinearity leading to variance inflation, models containing the school type as a covariate were not adjusted for age. Seasonal variations in purchasing behavior and behavioral differences between school lunch participants and non-participants were assessed using Pearson chi-squared tests, followed by respective post hoc analyses.

Study data were collected and managed using REDCap electronic data capture tools [35]. All analyses were conducted using R version 4.0 [36]. The level of significance was set to α = 0.05.

## 3. Results

### 3.1. Characteristics of the Study Sample

Out of 34 consenting schools, 3107 4th and 6th to 8th graders were eligible. Finally, 1215 students with 1037 parents took part in the study (Figure 1). The mean age of the study population was 11.32 years (SD = 1.35, range: 8.9–15.4), and around 50.8% were girls. Basic charactersitics of the study population are shown in Table 3. Participation rates varied by school type. In elementary/lower secondary/upper secondary schools, the participation rates were 43.6% (*n* = 690)/26.9% (*n* = 178)/47.2% (*n* = 347), respectively. The participation rate of elementary school parents was 88% (*n* = 607). In secondary schools, parental participation was 73.6% (*n* = 131) for lower secondary school and 86.2% (*n* = 299) for upper secondary schools.

### 3.2. School Lunch Participation

Table 4 shows the participation rates in school lunch in general and separately for the school types. Most of the students participated in school lunch. However, participation varied considerably between the school types, with the highest participation among elementary school students. Among the children studied, 3.5% (*n* = 42) usually bought something for lunch on the way or outside the school area, 27.4% (*n* = 333) usually brought packed lunch from home, and 66.3% (*n* = 805) usually participated in school lunch (multiple answers possible). Additionally, 16.6% (*n* = 202) usually eat nothing for lunch at school.

Associations of family-related predictors with school lunch participation are shown in Figure 2. Having breakfast on schooldays and high SES were positively associated with school lunch participation. In contrast, parental overweight, migration background, and many children were negatively associated with school lunch participation. Secondary school students, primarily from lower secondary schools, participated less frequently in school lunch than elementary school students. We did not find significant links between participation rates and BMI-SDS, FEB, or whether they purchase or bring something (lunch box, something to drink, fruits or vegetables).

Furthermore, we found a social gradient in school lunch participation. For all school types, the students from the high social strata participated more frequently, whereas socially disadvantaged students participated less frequently.

One-third of the students (32.8%; *n* = 394) did not participate in school lunch. As shown in Figure 3, the nonparticipating children stated most frequently that they did not like the taste, whereas a minority reported a special diet due to health reasons (e.g., allergies, food intolerances). Notably, the price was as decisive a reason for nonparticipation as the break time.

One-third of all parents reported being dissatisfied (31.7%; *n* = 300) with the food supply offered in school lunch. The proportion was larger in non-participants’ parents (63.7%; *n* = 158) than in participants’ parents (20.0%; *n* = 138). Figure 4 shows the reasons for parents’ dissatisfaction stratified by the participation status. The most common reason for parents’ dissatisfaction was that their children did not like the taste. Besides, parents were dissatisfied with the quality. Furthermore, parents of non-participants stated too high prices as a cause for dissatisfaction three times as often as participants’ parents (9.9% vs. 3.3%). A similar strong difference was only found for taste (34.8% vs. 12.7%).

### 3.3. Behavioral Differences among Participants and Non-Participants

Overall, the most frequent purchases were candy and chips (48.7%; *n* = 328) and ice cream (42.5%; *n* = 279) with a seasonal variation in purchasing ice cream (45.6% summer versus 38.4% winter, *p* = 0.077). Purchasing behavior differed between lunch participants and non-participants, particularly for sandwiches and energy drinks (Rates in Table 5). 

Almost all children (98.0%, *n* = 1182) brought something from home (lunch box, something to drink, a piece of fruit and/or vegetables) on at least three school days per week, with no significant difference between participants and non-participants. However, bringing a piece of fruit and/or vegetables was more frequent in participants than non-participants.

One-third of the parents stated that their child would be more likely to participate if school lunch was free of charge. The share was similar for participants (35.7%, *n* = 218) and non-participants (32.1%, *n* = 99). 

Only a few parents (5.3%, *n* = 64) reported that their child followed a special diet (vegetarian/vegan/gluten- or wheat-free/lactose-free) with no difference between participants and non-participants. An unfavorable family eating behavior was less common in participants than non-participants. While there was no difference in snacking behavior between participants and non-participants, not having dinner with the family and media usage during dinner were significantly common in non-participants.

### 3.4. Association of School Lunch Participation with Overweight

School lunch participants had a significantly lower BMI-SDS (β = −0.17, 95% CI: −0.31–−0.02, *p* = 0.019) and were significantly less likely to be overweight (OR = 0.66, 95% CI: 0.44–0.99, *p* = 0.043), but the associations lost statistical significance after adjustment (Table 6 and Table 7). A large part of the effect was explained by having breakfast on school days (BMI-SDS: β_adj_ = −0.57, 95% CI: −0.84–−0.30, *p* < 0.001; risk of overweight: OR_adj_ = 0.24, 95% CI: 0.13–0.47, *p* <0.001), which was highly correlated with lunch participation.

## 4. Discussion

We investigated predictors of school lunch participation and reasons for non-participation, i.e., possible intervention targets in 1215 German schoolchildren. The main finding of this study is that school lunch participation is primarily associated with family factors (migration background, parental overweight, SES, families with many children). The most stated reasons for nonparticipation were school-and lunch-related factors like taste, time constraints, and pricing. For children, time constraints were as important as pricing whereas parents reported pricing was as crucial a reason as the taste for nonparticipation. In line, one-third of the parents stated their child would be more likely to participate if school lunch was free of charge. Therefore, our data stress school-and lunch-related factors as an important opportunity to foster school lunch utilization.

### 4.1. School Lunch Participation and Determinants of Nonparticipation

More than two-thirds of the students reported participating in school lunch, a considerable higher percentage than the German average of 37.4% [6]. However, urban areas and regions in eastern Germany were shown to have higher participation rates. Therefore, our results can be considered representative for a city like Leipzig.

Participation rates varied considerably between school types, with the highest rates among elementary resp. lowest rates among lower secondary school students. These differences are confirmed by other studies [5,6,37] and can generally be explained by students gaining autonomy over their food choices as they get older, and thus deciding for or against school lunch participation. Lowest participation rates in lower secondary schools might reflect the higher proportions of low SES in these institutions as other surveys showed nonparticipation and meal skipping were more prevalent among low SES groups [38,39,40,41,42]. Indeed, regardless of the school type the participation was lower among children with low SES. Social inequality as a driver of nonparticipation was supported by the findings that parental overweight and migration background were also associated with lower school lunch participation. Both characteristics are also related to social inequalities [43]. Moreover, having more than two children was negatively associated with participation, suggesting that school lunch might be less affordable for families with more children, despite low-income families in Germany being eligible for free school lunch [44]. However, an application must be submitted for every child, a procedure that might be daunting, especially for low-educated parents. School-based lunch programs have the potential to address this problem and to facilitate access to lunch participation [2,6,22,45,46,47].

The most frequently reported reason for nonparticipation was the taste, which cannot be assessed objectively. As second most frequently reported reasons for nonparticipation, children stated the price and the break time (17% and 18%, respectively). Thus, we confirmed school- and lunch-related factors as critical in deciding for or against school lunch participation, as found in a German nationwide survey [6,37]. This finding agrees with qualitative and quantitative American research showing that students’ perceptions of food quality and school conditions, like long cafeteria lines and time constraints, are decisive regarding participation [19,21]. Finally, participants’ parents were three times more likely to be satisfied with the school lunch than non-participants’ parents, suggesting a strong dependence of parental perception regarding participation, which is in line with American research findings [20]. The authors found parental perception of the nutritional quality of school meals to be a significant predictor of participation, regardless of socioeconomic position. Furthermore, three times as many non-participants’ parents stated price or taste as a reason for dissatisfaction compared to participants’ parents. 

In summary, both parents’ as well as students’ dissatisfaction were mainly caused by lunch-and school-related factors. Since the taste is challenging to address, providing free school lunch at all schools might be an opportunity to boost school lunch utilization.

### 4.2. Behavioral Differences among Participants and Nonparticipants

Surprisingly, purchasing behavior was not significantly associated with school lunch participation. However, in general, we found non-participants purchasing something more often than participants. In particular, they bought sandwiches more often, presumably to cover their main meals. They also bought energy drinks more often, which is worrisome considering their excessive caffeine content and extraordinarily high sugar content. The finding is in line with higher consumption in German disadvantaged children [48]. However, these findings are contrary to a Norwegian study [49], where participants bought food from shops in the school environment more frequently than non-participants. These contrasting findings may originate from different settings and habits in other countries [50]. In the same study, nonparticipation was much higher (67%) and associated with higher SES. And indeed, for the US, the National Health and Examination Survey (NHANES) suggests that most elementary school students’ daily energy comes mainly from self-made purchases at stores, or quick- or full-service restaurants. School cafeterias contribute only a small percentage (5.5%) of the total energy [51].

Almost everyone brought something to drink, a lunch box, fruit, and vegetables from home. Thus, bringing something did not distinguish participants from non-participants. Since we did not check the actual components of the lunch boxes except for fruits/vegetables, it may contain energy-dense foods or sugar-sweetened beverages for some children. The fact that participants brought a piece of fruit and/or vegetables from home more often supports the assumption that they have overall healthier dietary habits, which needs further attention in future studies. 

If at least two unfavorable behaviors (out of three: no family dinner, media use during dinner, unhealthy snacking) were reported, the family eating behaviors were defined as unfavorable. It was more frequently observed among non-participants than participants. Moreover, breakfast skipping was also highly related to lunch-skipping. These finding supports the assumption that participation in school lunch reflects the general eating behavior and lifestyle factors, such as family meals and media use during mealtimes [52].

### 4.3. Association of School Lunch Participation with Overweight

The negative association between school lunch participation and overweight lost statistical significance after adjustment for other covariates. As we could show in previous research, having breakfast and lunch is associated with lower BMI-SDS and lower risk of being overweight [24]. This finding underlines the health-promoting effect of a high meal frequency and thus confirms existing literature [53,54,55]. Given that meal skipping is particularly prevalent among low SES and ethnic minority children [40], it is promising to target utilization through a family-independent intervention—thus, a school-based approach may help to reduce SES-induced differences in participation rates [23].

### 4.4. Limitations

We investigated various nutrition-related behaviors in a large sample of school children. However, some limitations should be acknowledged. Firstly, self-and parent-reported information (excluding anthropometry) might be subject to a social desirability bias, resulting in a systematic response error in the assessments of nutrition and health-related behaviors [56]. Secondly, parts of our questionnaires were self-designed after extensive literature review. Thus, the comparability is restricted. Further, the predefined response categories possibly caused bias in the results. Thirdly, there is an underrepresentation of disadvantaged/high-risk children. However, we would expect even stronger associations if participation rates of the high-risk population were higher. By choosing a school-based recruiting strategy, we tried to minimize this bias. Nonetheless, the representativeness, and thus the generalizability of the present study might be limited.

## 5. Conclusions

The results suggest that socioeconomic status and parental perceptions of school lunch play a key role in students’ participation. In general, school lunch participation was primarily associated with family-related factors (migration background, parental overweight, SES, families with many children). Most stated reasons for nonparticipation were school- and lunch-related factors like taste, pricing, and time constraints. Of note, one-third of the non-participants’ parents stated that their child would be more likely to participate if school lunch was free of charge. Therefore, our findings underscore the potential of community-based intervention strategies. Pricing and setting seem the most promising targets to fostering school lunch participation and its positive effects on child health.

## Figures and Tables

**Figure 1 nutrients-13-04209-f001:**
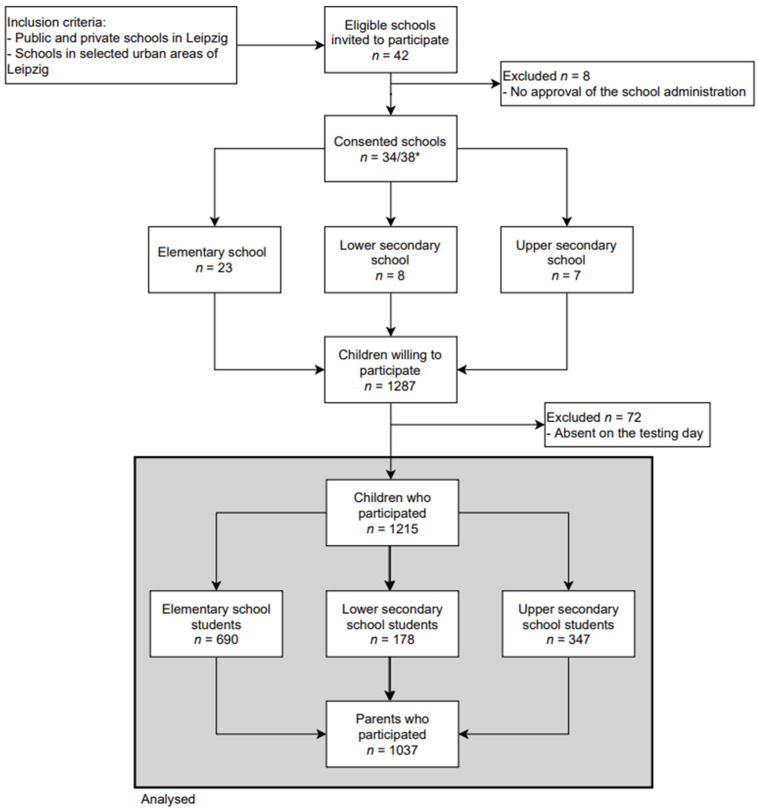
Flow diagram of inclusion and exclusion of study participants. * One school comprised an elementary school, a lower secondary school, and an upper secondary school; two schools comprised a primary school and an upper secondary school.

**Figure 2 nutrients-13-04209-f002:**
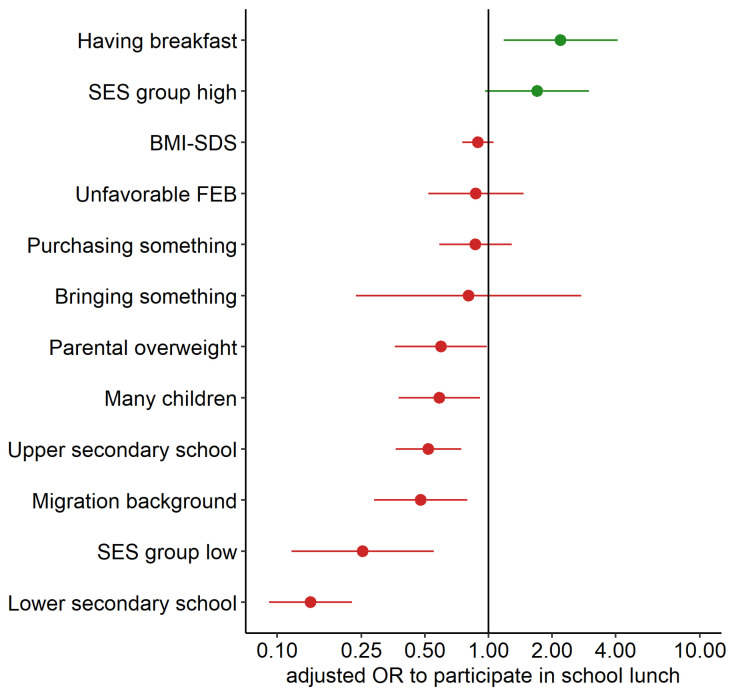
Associations between school lunch participation (*n* = 808) and different dependent variables, ordered by effect size and direction. Effects are given as adjusted odds ratios (incl. 95% CI), controlled for age and gender (School type adjusted only for gender). Positive effects are shown in green, and negative effects are shown in red.

**Figure 3 nutrients-13-04209-f003:**
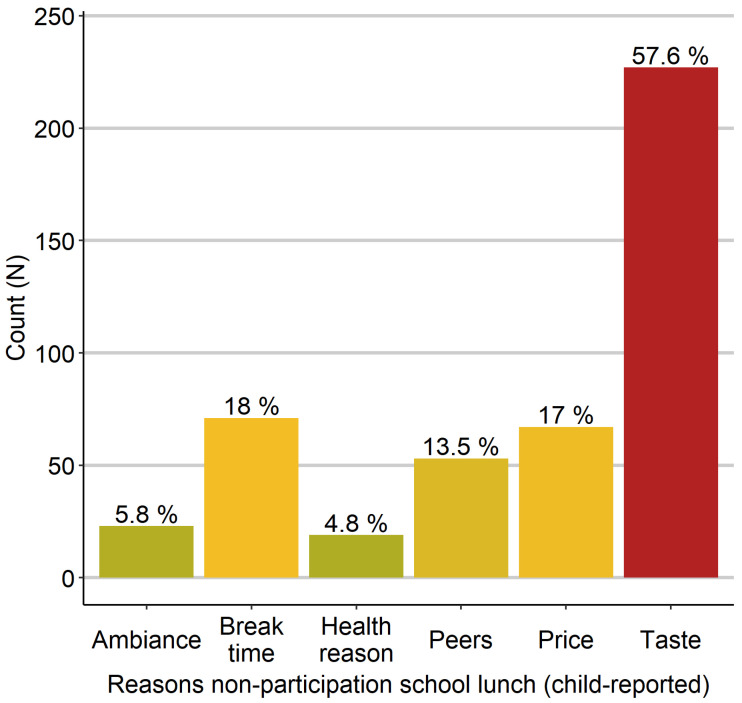
Absolute and relative frequencies of reasons for nonparticipation in school lunch (child-reported). More frequently reported reasons are shown in red, less frequently reported reasons are shown in green/yellow.

**Figure 4 nutrients-13-04209-f004:**
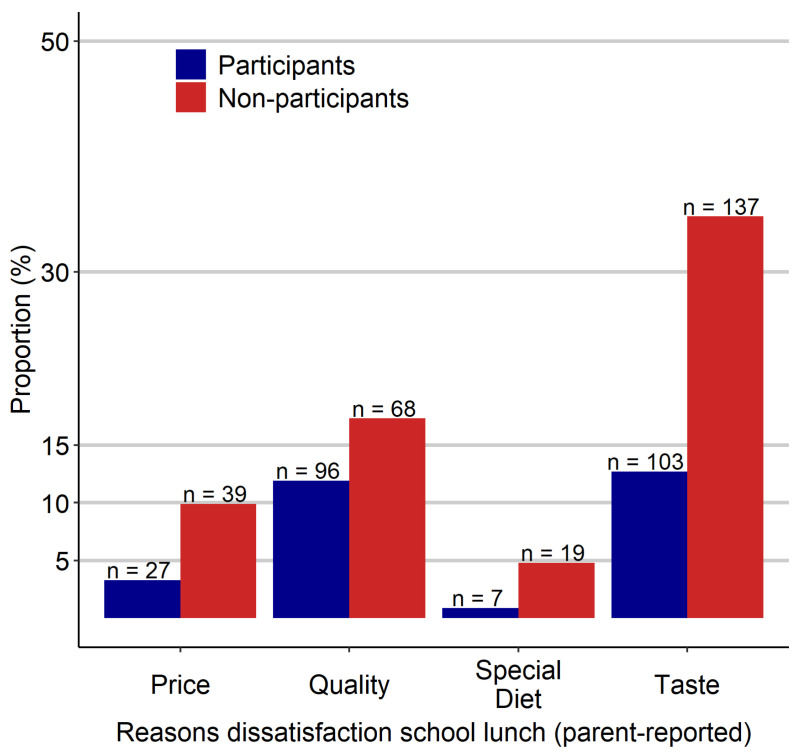
Absolute and relative frequencies of reasons for dissatisfaction with school lunch (parent-reported), stratified by students’ participation.

**Table 1 nutrients-13-04209-t001:** List of items regarding “eating behavior” (child questionnaire).

Question	Response Options
What do you usually eat for lunch at school?	
	I buy something on the way or outside the school area
	I eat what my parents give me
	I eat what is offered in the school canteen
	I eat nothing for lunch at school
Do you eat in the school canteen?	
	Yes
	No
Why not?	
	Because it’s too expensive for me
	Because I don’t like the taste
	Because I have allergies or have to follow a special diet
	Because my friends don’t eat there either
	I do not like the room (e.g., too loud, too dark)
	The break is too short to eat
Do you usually eat breakfast on school days?	
	No
	Yes, at home before I go to school.
	Yes, at school—I buy something
	Yes, at school—I bring something from home
Do you buy something more often than once a month…?	
	At a kiosk at school
	On the way to school
	During breaks outside school grounds
If yes, what do you buy?	
	Candy/chips
	Soda/cola/juice
	Energy drink
	Fruit/vegetables/salad
	Ice cream
	Sandwiches
Do you bring from home on three or more school days a week...?	
	A lunch box
	Something to drink
	A piece of fruit and/or vegetables from home

**Table 2 nutrients-13-04209-t002:** List of items regarding “eating behavior” (parent questionnaire).

Question	Response Options
I am satisfied with the school lunch.	
	Usually yes
	Usually no
If not, reasons for this are	
	The offer is too expensive
	The offer does not taste good to my child
	The quality of the offer is nonsatisfying (not age-appropriate, not healthy enough, not freshly prepared, or similar)
	The offer does not fit my child’s special needs (e.g., food intolerance, allergies, religion)
Would your child be more likely to use the school lunch if it was free?	
	Yes
	No
Does your child follow a special diet?	
	No
	Yes, vegetarian
	Yes, vegan
	Yes, gluten- or wheat-free
	Yes, lactose-free
Does your child usually eat dinner together with the family?	
	Yes
	No
Is the TV usually running at home during dinner, or is a tablet, smartphone, cell phone, or similar being used?	
	Yes
	No
Does your child usually snack between meals (e.g., chocolate, gummy bears, potato chips, pretzel sticks)?	
	Yes
	No

**Table 3 nutrients-13-04209-t003:** Description of study sample. Descriptives are given as mean and standard deviation for continuous variables and counts and percentages for categorical variables.

Study Population (*n* = 1215)	Mean (SD)	N
Age (years)	11.32 (1.35)	1215
	*n* (%)	
Gender		1215
Male	598 (49.2%)	
Female	617 (50.8%)	
School type		1215
Elementary	690 (56.8%)	
Lower secondary	178 (14.7%)	
Upper secondary	347 (28.6%)	
SES group		844
Low	90 (10.7%)	
Medium	467 (55.3%)	
High	287 (34.0%)	
Migration background		1000
Yes	196 (19.6%)	
No	804 (80.4%)	
	mean (SD)	
BMI-SDS	−0.04 (1.08)	1211
	*n* (%)	
BMI categorization		1211
Normal weight	1063 (87.8%)	
Overweight/obese	148 (12.2%)	
Parental overweight		894
Yes	618 (69.1%)	
No	276 (30.9%)	

*n*: count; SD: standard deviation; N: total; SES: socioeconomic status; BMI: body mass index; SDS: standard deviation score.

**Table 4 nutrients-13-04209-t004:** Proportion of students participating in school lunch, stratified by school type.

	Elementary School	Lower Secondary School	Upper Secondary School	Total
School lunch participants	526	59	223	808
Nonparticipants	152	118	124	394
Total	690	178	347	1215
Participation rate (%)	77.6	33.3	64.3	66.5

**Table 5 nutrients-13-04209-t005:** Percentage of students by participation status and different behavior. Pearson chi-squared test was used to compare intergroup differences.

	School Lunch Participants *n* (%)	Nonparticipants *n* (%)	*p*-Value
Purchasing something	314 (40.0)	197 (51.0)	<0.001
Sandwiches	144 (36.2)	132 (48.9)	0.001
Energy drinks	23 (6.2)	46 (18.6)	<0.001
Candy/chips	185 (47.1)	141 (51.1)	0.345
Soda/cola/juice	130 (33.7)	104 (39.1)	0.182
Ice cream	175 (44.6)	101 (38.9)	0.166
Fruit/vegetables/salad	94 (24.9)	74 (29.5)	0.235
Bringing something	791 (98.3)	381 (97.4)	0.467
Lunch box	723 (90.9)	335 (87.7)	0.104
Something to drink	768 (96.4)	366 (94.1)	0.100
Fruit and/or vegetables	656 (84.0)	277 (74.7)	<0.001
Unfavorable FEB	108 (15.4)	71 (22.6)	0.007
Unhealthy snacking	405 (59.6)	196 (63.2)	0.305
Dinner without family	15 (2.1)	18 (5.8)	0.005
Media usage during dinner	137 (19.5)	89 (28.4)	0.002
Total	808	394	

*n*: count; FEB: family eating behavior.

**Table 6 nutrients-13-04209-t006:** Associations between BMI (*n* = 808) and school lunch participation. Effects are given as β (95% CI).

		BMI-SDS					
		Unadjusted Analysis			Adjusted Analysis		
		β	95% CI	*p*-Value	β_adj_	95% CI	*p*-Value
School lunch participation	Participants	−0.17	(−0.31–0.02)	0.019	−0.11	(−0.27–0.05)	0.165
	Nonparticipants	Ref.					
Having breakfast	No	1.11	(0.93–1.31)	0.096	−0.57	(−0.84–−0.30)	<0.001
	Yes	Ref.					
Parental overweight	No	0.64	(0.52–0.80)	0.070	0.57	(0.42–0.72)	<0.001
	Yes	Ref.					

β: unadjusted effect; β_adj_: adjusted for parental overweight and having breakfast.

**Table 7 nutrients-13-04209-t007:** Associations between risk of being overweight (*n* = 808) and school lunch participation. Effects are given as odds ratios (95% CI).

		Overweight					
		Unadjusted OR			Adjusted OR		
		OR	95% CI	*p*-Value	OR_adj_	95% CI	*p*-Value
School lunch participation	Participants	0.66	(0.44–0.99)	0.043	0.78	(0.47–1.29)	0.322
	Nonparticipants	Ref					
Having breakfast	No	Ref					
	Yes	0.32	(0.20–0.52)	<0.001	0.24	(0.13–0.47)	<0.001
Parental overweight	No	Ref					
	Yes	4.18	(2.13–8.20)	<0.001	4.31	(2.14–8.68)	<0.001

OR: odds ratio; OR_adj_: adjusted for parental overweight and having breakfast.

## Data Availability

The legal requirements and the given informed consent do not allow public sharing of the dataset. Interested researchers can contact the research data management of the Medical Faculty, University Leipzig: forschungsdaten@medizin.uni-leipzig.de for further information. The dataset ID is PID-00080/01.

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
