# Peer review of "Purchasing Behavior, Setting, Pricing, Family: Determinants of School Lunch Participation"

_nutrients, 2021, doi:10.3390/nu13124209_

Round 1

Reviewer 1 Report

Materials and Methods

Regarding the study variables, have the authors taken into account the existence of a validated questionnaire? The questions used, have they been designed for the study? If so, explain the method followed to define them (group of experts, literature review, etc.).
The questions used to collect eating habits and their possible answers could be collected in a table, this would simplify the Data section.

In the Statistical Analysis section, a hierarchical logistic regression is mentioned for the dependent variables overweight / obesity, and for the BMI-SDS, with school as a random effect. However, in the results section, the results shown correspond to a multivariate model of linear regression (for BMI) and another of logistic regression (for overweight / obesity).The authors must modify either the wording of the statistical analysis or the results section, indicating what was found with the hierarchical model.

Results

The authors should better describe the study population at the beginning of the results section. Not only describe the number of participating students and parents, but also include what is the age range, and the percentage of participants by gender.

In table 1, replace μ by mean or , since in statistical inference, μ is used to refer to the population mean, and not to the sample, as is this case.

Figure 3 does not provide relevant information, it can be eliminated and refer to these results only in the text. 
In figure 5, it is recommended that absolute frequency (n) appear together with relative frequency (%). 
In Table 3, absolute frequencies should appear together with relative frequencies, as well as the total of participants and non-participants in the table header. 
Table 4 must be modified. Coefficients b appear for the BMI-SDS, which, being a numerical variable, must refer to the coefficients of a linear regression model, which has not been specified in the statistical analysis section. The crude ORs from the univariate regression analysis should be added for the rest of the independent variables, in both models (BMI-SDS and everweight / obese). It would be better if the models for the two dependent variables are expressed in separate tables, and in each of them show the results of the univariate and multivariate model. Also, at the bottom of table 4, βadj and ORadj are shown, but this does not appear in the table.

Reviewer 2 Report

Thank you for the opportunity to review this manuscript. My specific comments are below:

  • Please clarify what 'all day offers' means for readers not familiar with the country/context (line 39)
  • Why are the numbers of all day schools presented as a proportion with decimal points?
  • (Line 43) I am not sure how the education day has extended if the teaching hours are unchanged. Please clarify. 
  • Line 44 is a significant statement, with only one reference. 
  • Line 47 appears that there is a word missing
  • Line 53 change 'the' to 'all'
  • Lines 56/57 are a bit confusing, they may need to be separated?
  • Lines 61 - 74 seem a little out of context. This section could flow better from the previous paragraphs. There needs to be more rationale for the study included.

Methods

  • How was parental consent obtained? Was this written or verbal?
  • Did the child get the incentive before/after parental consent?
  • Line 92 is not clear
  • Line 123, were these pre-defined responses?
  • Line 136, follow, not follows
  • Line 140 options are all 'treat type' foods, did you capture fruit or other healthier foods as a snack?
  • Line 142, why was this measure chosen as 'unfavourable', was this based on other tools? This needs justification. 
  • Line 178: apori?

Results and discussion

  • Table 1. Is there a way to reformat this so that there is not so much white space in the second column. Perhaps this could be added as a footnote.
  • Figures, not sure why some are B/W while figure 5 is coloured.
  • Line 276, needs review for grammar
  • Some paragraphs are very long and discuss more than one point. Suggest revisting this and reviewing paragraphing structure. 
  • Line 388 reads emotively, please include a reference if it stays as currently is

Reference 47 is incorrect

Please also consider providing more critical thought in regards to the questions used and interpretation of these (please also see above). 
